# Continuous Circulation of Yellow Fever among Rural Populations in the Central African Republic

**DOI:** 10.3390/v14092014

**Published:** 2022-09-12

**Authors:** Huguette SIMO TCHETGNA, Stéphane DESCORPS-DECLERE, Benjamin SELEKON, Sandra GARBA-OUANGOLE, Xavier KONAMNA, Mathieu SOUNGOUZA, Gaspard TEKPA, Pierre SOMSE, Emmanuel NAKOUNE, Nicolas BERTHET

**Affiliations:** 1Centre for Research in Infectious Diseases, Yaoundé P.O. Box 13591, Cameroon; 2Institut Pasteur, Université Paris Cité, Bioinformatics and Biostatistics Hub, 75006 Paris, France; 3Institut Pasteur de Bangui, Bangui P.O. Box 923, Central African Republic; 4Centre de Santé de Mboki, Mboki, Central African Republic; 5Service d’Infectiologie, Hôpital de l’Amitié, Bangui, Central African Republic; 6Ministry of Health, Bangui P.O. Box 883, Central African Republic; 7Cellule d’Intervention Biologique d’Urgence, Unité Environnement et Risque Infectieux, Institut Pasteur, 75015 Paris, France; 8Unit of Discovery and Molecular Characterization of Pathogens, Centre for Microbes, Development, and Health, Institut Pasteur of Shanghai, Chinese Academy of Sciences, Shanghai 200031, China

**Keywords:** haemorrhagic fever, next-generation sequencing, Africa, remote settings, yellow fever

## Abstract

Yellow fever remains a public-health threat in remote regions of Africa. Here, we report the identification and genetic characterisation of one yellow-fever case observed during the investigation of a cluster of nine suspected haemorrhagic fever cases in a village in the Central African Republic. Samples were tested using real-time RT-PCR targeting the main African haemorrhagic fever viruses. Following negative results, we attempted virus isolation on VERO E6 cells and new-born mice and rescreened the samples using rRT-PCR. The whole viral genome was sequenced using an Illumina NovaSeq 6000 sequencer. Yellow-fever virus (YFV) was isolated from one woman who reported farming activities in a forest setting several days before disease onset. Phylogenetic analysis shows that this strain belongs to the East–Central African YFV genotype, with an estimated emergence some 63 years ago. Finally, five unique amino-acid changes are present in the capsid, envelop, NS1A, NS3, and NS4B proteins. More efforts are required to control yellow-fever re-emergence in resource-limited settings.

## 1. Introduction

Yellow fever (YF) is one of the most severe mosquito-borne viral diseases in the tropical regions of the world, responsible for 84,000–170,000 severe cases and nearly 29,000–60,000 deaths annually in South America and Africa [1]. An estimated 400 million unvaccinated people are at risk of infection despite the availability of a safe and effective vaccine for over eight decades [2]. Models predict that mass-vaccination activities in Africa can reduce YF deaths by 47% (CI 95% [10–77%]) [3]. The genome of the causative agent (yellow-fever virus (YFV), genus *Flavivirus*, family *Flaviviridae*) is a single-stranded, positive-sense RNA molecule of ~11 kb that encodes a single polyprotein that matures into three structural proteins (C, prM/M, E) and seven non-structural proteins (NS1, NS2A, NS2B, NS3, NS4B, NS4B, NS5) [4]. YFV corresponds to a unique serotype further classified into seven geographically segregated genotypes, including five actives in Africa (East African, East–Central African, Angolan, West African I and II) and two in South America (South American I and II) [5,6].

YFV is a zoonotic virus that usually evolves in a sylvatic transmission cycle between non-human primates and various canopy mosquitoes such as *Aedes* spp., *Haemagogus* spp., and *Sabethes* spp. [7]. Humans become infected in forest settings during occupational or recreational activities and tourism. The urban transmission cycle of YFV is rare, but can potentially be catastrophic, as seen during the outbreak in Angola and the Democratic Republic of the Congo (DRC) in 2015–2016 [8], prompting the vaccination of more than 30 million people in the affected locations. Most YF cases go unnoticed, but symptomatic infections include fever, headache, nausea, muscle pain, backache, vomiting, jaundice, and minor mouth, nose, and eye bleedings [7,9]. Around 20 to 60% of YF cases evolve into a more severe disease characterised by haemorrhage, renal failure, and hepatitis due to multiple-organ failure and may lead to death within 7–10 days after the onset of symptoms [9,10,11,12].

Like many other African countries, the Central African Republic (CAR) is known to be endemic to YF. Since the first laboratory-confirmed YF case in the 1930s, only 11 other cases have been registered during the following 60 years in the CAR [13,14]. Nevertheless, an increase in YF records was observed in 2008 and 2009 [15], marking an important turning point in YF epidemiology for this country. As such, YF surveillance from 2007 to 2012 enrolled approximately 3220 suspected YF cases, with 55 confirmed cases and 11 deaths [16]. Additionally, during the same period, almost 13% of the CAR’s population was carrying naturally acquired antibodies against YFV [14]. Entomological surveys have reported a wide diversity of YFV vectors in the CAR, with, however, a disparity in their geographic distribution [17]. Nonetheless, these surveys have resulted in the detection of over 40 YFV strains from a wide variety of *Aedes* mosquitoes, including *Aedes opok* and *Ae africanus*, across the country.

Here, we report the case of a woman who developed YF after returning home from farming activities in a gallery forest in the southeastern part of the CAR.

## 2. Materials and Methods

### 2.1. Virus Detection

In August 2018, a team of field epidemiologists was dispatched to the village of Mboki, Haut Mbomou prefecture (5°18′57″ N, 25°57′29″ E), in the southeastern region of the CAR, near the DRC border, to investigate a cluster of suspected haemorrhagic fever cases (Figure 1). Therefore, nine blood samples were collected and transported to the Institut Pasteur de Bangui virology laboratory for analysis. Total RNA was extracted from plasma using the QIAamp^®^ Viral RNA mini kit (Qiagen, Hilden, Germany) and retrotranscribed using the High-Capacity cDNA Synthesis kit (Applied Biosystems, Foster City, CA, USA). Ebola, Marburg, and yellow-fever viruses were screened using real-time RT-PCR with the RealStar Filovirus Screen RT-PCR Kit v1.0 and RealStar Yellow Fever RT-PCR Kit v1.0, respectively (Altona Diagnostics, Hamburg, Germany). Additionally, Rift Valley fever, yellow-fever, and Crimean–Congo haemorrhagic fever viruses were screened using real-time RT-PCR with the TaqMan Universal PCR Master Mix kit (Applied Biosystems, Foster City, CA, USA) and primers published elsewhere [18,19,20]. Simultaneously, the plasma samples were diluted 1:10 with 1X PBS and used to attempt virus isolation on suckling-mouse brain. The isolation was evaluated through the recording of lethargy, bristle fur, paralysis, and death. After two passages, the virus was propagated once on VERO E6 cells, and a cytopathogenic effect was observed. Then, a second run of real-time RT-PCR tests was performed as described above.

### 2.2. Whole-Genome Sequencing

Total RNA was extracted from cell-culture supernatants using the QIAamp^®^ Viral RNA mini kit (Qiagen, Hilden, Germany) and used for the synthesis of the first cDNA strand using SuperScript III reverse transcriptase and random primer hexamers (Invitrogen, Life Technologies, Carlsbad, CA, USA). The second cDNA strand was obtained using the NEBNext^®^ Ultra™ II Non-Directional RNA Second Strand Synthesis Module kit (New England BioLabs, Hitchin, UK). Library preparation was then completed with the NEBNext^®^ Ultra™ II RNA Library Prep Kit for Illumina (New England BioLabs). After quality validation on the 2100 BioAnalyzer (Agilent Technologies, Santa Clara, CA, USA), the library was sequenced to obtain 2 × 150 bp paired-end reads on an Illumina NovaSeq 6000 sequencer (Integragen, Ivry, France).

### 2.3. Bioinformatics and Phylogenetic Analysis

After quality control and trimming, the reads were assembled de novo with SPAdes v3.7 (Waltham, MA, USA) [21] to produce a 10,830 bp fragment representing the whole genome of YFV (Genbank accession No. MW960207). A multiple-sequence alignment was performed with YFV reference genomes available in GenBank using MAFFT v7 in Unipro Ugene version 34.0 (Novosibirsk, Russia) [22] and manually edited. Then, a phylogenetic tree was inferred on the whole genome using the maximum-likelihood (ML) method implemented in IQ-Tree version 1.6.12 (Vienna, Austria) using the best substitution model as determined by the same software (GTR + F + R2) [23,24]. The node supports were estimated from 1000 bootstrap replicates. Furthermore, to gain more insight into the evolution of YFV and because only a few YFV whole genomes are available in Africa, we also performed an ML phylogenetic analysis on the envelope gene using the appropriate substitution model (TIM2e + I + G4). A pairwise comparison of the YFV CAR strain to its closest relatives from Uganda and China (imported from Angola) at the amino-acid level was performed. Finally, BEAST version 1.10.4 (Auckland, New Zealand) was employed to infer both the YVF phylogenetic tree and the associated chronogram. BEAUti (Bayesian evolutionary analysis) software, bundled with the BEAST software suite, was first used to instantiate the evolutionary model and MCMC parameters. We select a relaxed (uncorrelated lognormal) molecular clock and the most general substitution model (GTR + I + G4) as the evolutionary process description. We also employed the Bayesian skyline population coalescent prior as it is the most frequently used descriptor of the complex population dynamics of YFV in the literature [25]. A large MCMC chain of 10,000,000 iterations was set with a sampling frequency of 1000 to ensure stationarity and convergence. Completed BEAST runs were checked visually for convergence and sufficient sampling of the posterior space (ESS > 200) with Tracer version 1.7.1 (Auckland, New Zealand). Finally, the final chronogram was generated using TreeAnnotator version 1.10.4.23 (Auckland, New Zealand) [26].

## 3. Results

### 3.1. Case Description

In August 2018, a cluster of nine people was investigated for suspected haemorrhagic fever in the village of Mboki, Haut-Mboumou prefecture. The real-time RT-PCRs performed directly on the primary plasma samples for the detection of the main African haemorrhagic fever viruses were negative. Upon multiplication in new-born mice and Vero E6 cells, YFV was the only virus identified in one out of the nine samples. Threshold-cycle values of 14.36 and 16.81 were obtained for new-born-mouse brain-tissue homogenates and cell-culture supernatants, respectively. The YFV-positive sample was from a 50-year-old woman who presented with fever (axillary temperature above 38.5 °C), headache, arthralgia, cough, vomiting, nausea, and abdominal pain, but no haemorrhagic or neurologic impairment at the time of inclusion. She reported farming activities in a gallery forest before the onset of the disease but did not recall having ever received the YF vaccine. Fortunately, this woman survived the YF infection.

### 3.2. Molecular Analysis of the YFV Strain Identified

We studied the genomic characteristics of the newly isolated YFV strain for more insights into its evolution. We therefore compared the YFV genome obtained with other African YFV sequences retrieved from databases. At the nucleic acid level, genetic divergence ranged from 2% to 21% between this new CAR strain (YF-118) and YFV strains from East and West Africa, respectively. Its closest relatives were reported from Uganda (two strains) and Ethiopia with genetic divergences of 2%, 4%, and 7%, respectively (Genbank accession Nos. JN620362, DQ235229 and AY968065), and up to 21% divergence was observed with the YFV prototype strain Asibi (Ghana, 1927) (Genbank accession No. AY640589). The amino-acid level showed a similar trend, with almost 7% difference with the Asibi strain and other strains from West Africa and less than 1% difference with strains from Uganda, as described elsewhere [27]. We observed 12, 106, and 227 amino-acid substitutions between the CAR YF-118 strain and strains from Uganda 2010, Shanghai (imported from Angola in 2016), and Asibi, respectively (Figure 2). Five unique mutations were observed in the capsid (68F), envelop (371K), NS2A (1319R), NS3 (1869R), and NS4B (2503E), and an additional 26 unique mutations were specific to our strain and Uganda, probably because they belong to the same genotype (Figure 2). Additionally, the YF-118 sequence is almost identical to other strains from the CAR isolated in 1977, 1985, and 1986, with which it shares 98–100% similarity at the nucleic-acid level for the envelope protein (Genbank accession Nos. U23573, U23571, AY839634), although these strains were isolated 30 to 40 years prior, either from humans or mosquitoes.

The phylogenetic analysis of the whole YFV genome is shown in Figure 3a. As expected, the YFV genotypes displayed a strong geographic association. Our YFV strain clustered with YFV strains from the same sub-region, within the East–Central African genotype. Accordingly, it formed a distinct clade with strains from Uganda and Ethiopia and was closely related to the strain from Uganda 2010, as already observed in the pairwise comparison. This clade was distinct from the Angolan clade, which contains strains from the last 2015–2016 YFV outbreak in that country. Interestingly, the envelope-protein phylogeny provides more information on YFV evolution (Figure 3b). The East–Central African genotype seems to be subdivided in two clusters, one with strains from the CAR and Uganda and the other with strains from Uganda, Ethiopia, and Sudan. The strain from Uganda 2010 shares a common ancestor with all the YFV strains from the CAR, whereas YF-118 seems to have evolved from a common ancestor with the CAR strains from 1977 and 1986.

This observation agrees with the results of the molecular-clock analysis (Figure 4), which shows that the most recent common ancestor (TMRCA) of YF-118 and two other CAR YFV strains appeared around year 1955 (95% highest posterior density (HPD) 1937–1966), some 63 years ago, whereas this clade diverged from the clade of Uganda 2010 around year 1918 (95%HPD 1882–1952), implying a recent evolution of YFV in the CAR. These estimates are compatible with the predicted emergence of the East–Central African genotype, some 250 years ago [25,27], and the slower evolutionary rate of YF viruses, especially compared with other flaviviruses, including dengue viruses [28].

## 4. Discussion

In the present article, we described a yellow-fever case observed in August 2018 in Mboki, a village of the southeastern forested region of the CAR. The case was discovered during the investigation of a cluster of nine individuals for a possible haemorrhagic fever epidemic (the patient survived). No further epidemiologic investigation was carried out to assess the extent of this YFV outbreak due to logistical issues. Additionally, no YFV vaccination had been provided to this remote population for almost eight years before this YF occurrence. We postulate that YF cases are recurrent in this area, although no other YF case was recorded in 2018 (Nakoune, personal communication). Indeed, most YF outbreaks in Africa and Latin America are reported from rural areas, linked with the disruption of the YFV sylvatic transmission cycle due to tourism and occupational or recreational activities in the forest [7,29,30]. Accordingly, in our CAR case, the gallery-forest setting of this semi-rural location means that it hosts a wide variety of non-human primates, and the local population heavily frequents the forest, relying on it for food and firewood. Moreover, we suppose that the samples were not taken at the peak of the outbreak, but rather later, hence the apparent low viral load of the sample, which required a prior virus isolation step before detection with real-time RT-PCR. A low viral load may also explain why YFV was detected only in one of the nine suspected cases. Delays in case notification and identification are major obstacles in the investigation and containment of emerging viruses in remote regions [31,32]. Moreover, the method used in this case was very time-consuming and may not be appropriate for dealing with a suspected outbreak. More sensitive and straightforward methods are highly recommended.

YFV genetic diversity and evolution are not yet fully understood, mostly due to the non-availability of genomic sequences and the disparity in representatives from some geographic regions and countries, especially Central and East Africa [33]. Here, we describe a new YFV sequence from the CAR, belonging to the East–Central African genotype, and therefore contribute to improving the knowledge of YFV evolution in this region. Our findings are consistent with the criteria of YFV genotype classifications and genetic divergence previously observed among African YFV strains [5,6,34]. Previous reports indicate genetic diversity of 10–23% at the nucleic-acid level for African YFV genotypes [6,27], with the highest diversity occurring between the East and East–Central genotypes compared with West African I or II; the maximum amino-acid sequence divergence between these genotypes is ~8% [4,34]. Most of the amino-acid changes observed in our strain have already been described elsewhere [27,34]. These mutations have not been associated with any major phenotypic variation, thus explaining the continuing effectiveness of the YFV vaccine against all genotypes over time [4,25,34]. We observed five unique amino-acid changes located in the capsid, envelop, NS2A, NS3, and NS4B, three of which are involved in genomic RNA replication. Whether these specific amino-acid changes are specific to this strain remains to be confirmed [35,36]. Some of them, if not all, may occur in unidentified strains that have not yet been sequenced. This issue will be well elucidated by the analysis of other circulating YFV genomes from infected humans and mosquitoes in the CAR.

Overall, the topologies of the phylogenetic trees built either with the whole genome or only with the envelope gene were similar, suggesting that the evolutionary rate of YFV is similar across the genome. Such a pattern has also been observed when comparing the PreM/E junction, the envelope gene, and the whole genome [4,37], with a clear geographic specialisation of the genotypes. Although there is some gene flow between regions, YFV evolution can occur in situ in Africa, consistent with the restricted mobility of the non-human primate reservoirs and the canopy mosquitoes involved in the sylvatic transmission [28]. This restricted gene flow implies that our YFV strain may have been active in that location for many years among non-human primates and that the case observed was the result of a spill-over event. Nonetheless, this border region is characterised by the presence of Congolese and South Sudanese refugees, as well as foreign humanitarian or migrant workers, hence increasing the risk of importation of the virus to other regions. Indeed, YF importation has been observed in Europe and Asia, mainly associated with the low vaccination coverage of travellers. For instance, Chinese workers who imported YF to China from Angola during the 2015–2016 outbreak were not vaccinated against YFV [38], a failure of the international-health regulation regarding travel to yellow-fever-endemic regions [38,39].

The molecular-clock analysis performed on this new strain identified the MRCA of our strain around year 1955 (some 63 years ago), although it was detected only in 2018. The molecular clock also suggested that our strain is older than those detected in 1977 and 1986 among humans and mosquitoes in the CAR. This result provides support for the idea that YFV has been active for a very long time in the CAR and that the virus is continuously circulating in the southeastern region of the country, likely among unidentified hosts.

Interestingly, in the CAR, at least two YFV genotypes have been active, as shown in the envelope-gene ML tree (East–Central genotype and West African II genotype) (Figure 3). The circulation of two YFV genotypes has previously been suggested in different studies, both in the CAR and in Uganda, and is in favour of a Central African origin of YFV, followed by dispersal to West Africa and Latin America [28,34,40]. However, only a few human cases or small outbreaks have ever been reported from East and Central Africa despite the high genetic diversity observed locally [27]. Many hypotheses have been suggested to explain these discrepancies, with the most plausible being the reduced contact between the human population and vectors, the transmission efficiency, and the cross-immunisation of the population by other flaviviruses active in the area (reviewed in [33]). Unlike in West Africa and Latin America, where most YF studies are carried out, little is known about YFV transmission in Central Africa. In general, in Central Africa, little is known about the distribution and bioecology of the sylvan vector and there is a lack of vector-competence studies, which affects our ability to predict potential outbreaks. Around 20 species of *Aedes* mosquitoes have been identified in the CAR, among which are *Ae. Aegypti, Ae. Africanus, Ae. Simpsoni, Ae. Luteocephalus, Ae. Vittatus*, and *Ae. Opok* [17], which are known vectors of YFV. However, to the best of our knowledge, the only vector-competence study performed in this African sub-region focused on urban *Ae. aegypti* and *Ae. albopictus* and showed contrasting results depending on the mosquito population [41]. Thus, many knowledge gaps remain, hindering our understanding of the transmission, maintenance, and emergence of YF in Central Africa and therefore the control of the disease.

## 5. Conclusions

Despite many efforts, YF remains a public health issue in the CAR. The control of this disease will continue to be a challenge, especially in regions where the population relies heavily on the forest for their survival, thereby disrupting the sylvatic transmission cycle. It is thus urgent to improve the vaccination coverage through mass campaigns and childhood immunisation to prevent YFV from spreading in remote villages. A consequence of the early development of the YF vaccine is the lack of basic information on the biology of this virus. As a result, the genetic diversity of the virus and its role in virus virulence is not well understood, especially for African genotypes of YFV, and calls for more studies like this one.

## Figures and Tables

**Figure 1 viruses-14-02014-f001:**
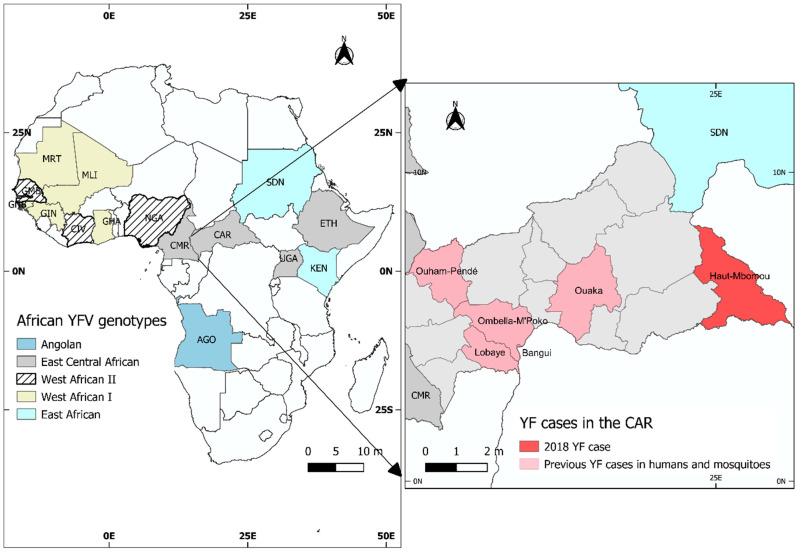
Origin of the African yellow-fever virus (YFV) sequences used in this study. The left panel shows the geographic distribution of YFV genotypes in Africa. The right panel shows the distribution of YFV cases in the CAR, including the region of origin of the case described in this study.

**Figure 2 viruses-14-02014-f002:**
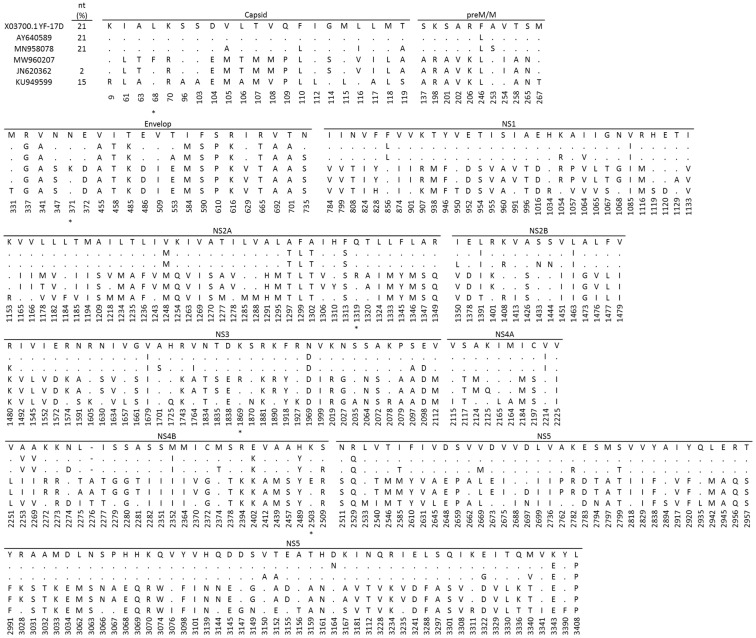
Pairwise comparison of the full-length yellow-fever virus polyprotein from CAR. Amino-acid mutations are represented by the one-letter code and conserved residues are represented by dots. Unique amino-acid changes present in the studied strain are shown by an asterisk (*). nt (%) represent the distance between our sequence and the other strain. X03700.1 YF-17D yellow-fever vaccine strain, MW960207 is the strain described in this study.

**Figure 3 viruses-14-02014-f003:**
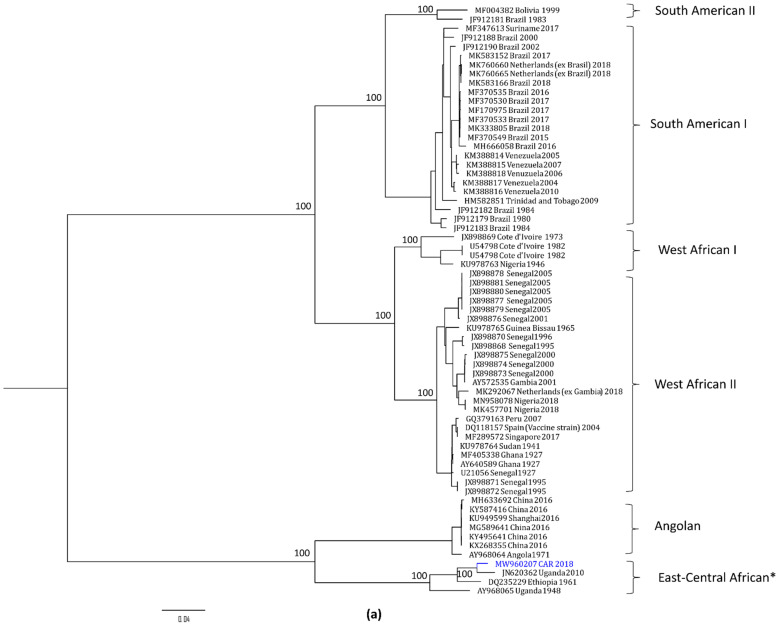
Maximum-likelihood phylogenetic tree of the yellow-fever virus. The figure shows the maximum-likelihood trees built using IqTree v1.6.12 software based on the whole yellow-fever virus (YFV) genome (panel (**a**), GTR + F + R2 substitution model, 10,548 bp) or the envelope gene only (panel (**b**), TIM2e + I+G4 substitution model, 1455 bp). The strain described in this study is shown in blue. In panel (**a**), East African and East–Central African genotypes, indicated by an asterisk (*), cluster together artificially likely due to the lack of available full-length sequences.

**Figure 4 viruses-14-02014-f004:**
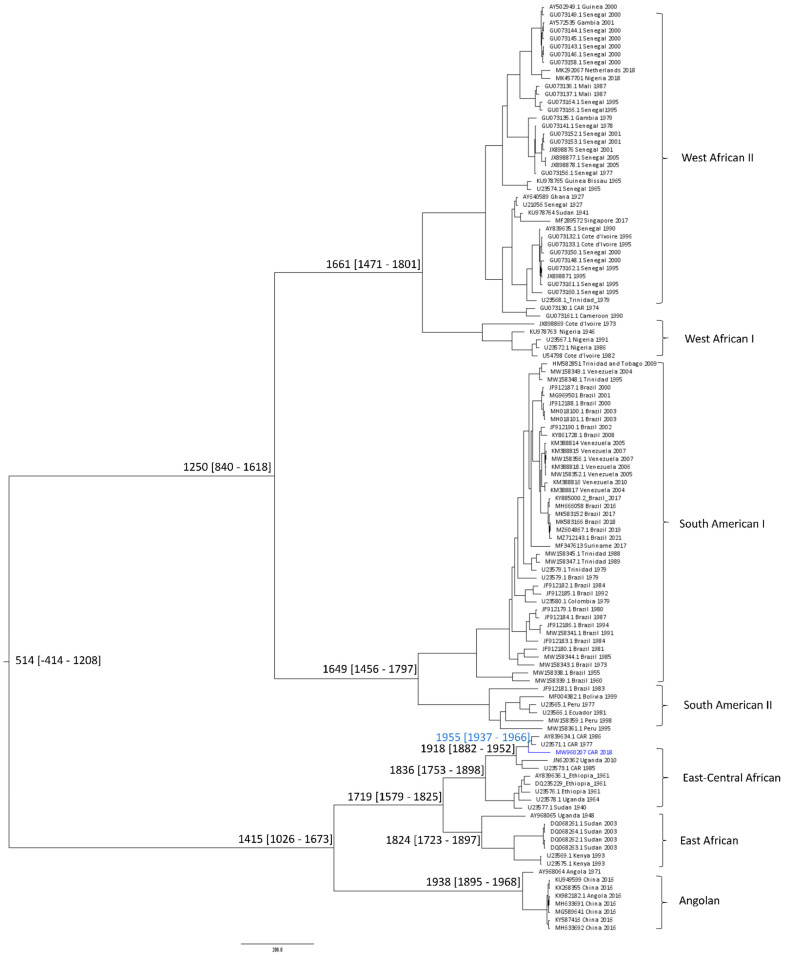
Bayesian chronogram of yellow-fever virus (YFV). The Bayesian analysis was performed on the envelope gene (1455 bp). The YFV genome isolated in the Central African Republic is indicated in blue. Median divergence times are indicated at the major nodes and the 95% highest posterior density (HPD) is given in brackets. TMRCA is show in blue.

## Data Availability

The yellow-fever virus sequence obtained during this study has been deposited in GenBank under the accession number MW960207.

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
