# Peer review of "Continuous Circulation of Yellow Fever among Rural Populations in the Central African Republic"

_viruses, 2022, doi:10.3390/v14092014_

Round 1

Reviewer 1 Report

An interesting paper but there are ways to make the paper stronger.

1) This is a molecular paper and there should be more and better description pertaining to the protein biology of flaviviruses/YFV. The authors can use any references or virology textbooks:

https://www.sciencedirect.com/science/article/abs/pii/S1879625712000387

3) I think the authors did not take full advantage of the knowledge and tools available in modern protein biology to address many questions. The protein biology of flaviviruses is important as specific proteins are linked to vaccine/antibody evasions and virulence respectively.

https://www.ncbi.nlm.nih.gov/pmc/articles/PMC2395160/

https://pubmed.ncbi.nlm.nih.gov/31698857/

https://pubmed.ncbi.nlm.nih.gov/27102744/

4) As you can see in the literature above, there are ways to extrapolate the levels of virulence and vaccine/antibody evasion. For example, we could extrapolate the chances of vaccine evasion by doing a sequence similarity comparison of the enveolope proteins of variant in question to the ones used in the vaccine.

5) Do the authros have data pertaining to the virulence of the various variants?

https://bmcinfectdis.biomedcentral.com/articles/10.1186/s12879-021-06535-4

Author Response

Thank you for these comments. I will respond to all five preoccupations together as they are all oriented toward amino acid and their function. Amino acids of the described viral strain have been thoroughly studied (see figure 2, page 5) 5 unique amino acid have been observed in the Central African Republic (CAR) strain, in the envelop, capsid, NS2A, NS3 and NS4B. The mutation observed in the envelop is not available in the other envelop sequences available for CAR but there not means to test if this mutation is fixed in CAR. Since our sequence is the first whole genome available in the country, we cannot test the other unique mutations. These changes may influence vaccine evasion and the virus multiplication, but since we only have one sequence, we cannot conclude on this matter. Further studies are needed to first evaluate if these amino acid changes are conserved in the country and then to assess their role in viral virulence, immune escape and vaccine response. (see line 255-261)

Reviewer 2 Report

The manuscript "Continuous circulation of yellow fever among rural populations in the Central African Republic" present a case of isolation and genetic description of a YFV strain in 2018. The work is nicely performed and presented. Nevertheless, the manuscript describes only one strain and its sequence analysis, which, in my opinion, is not enough for a full-type Original Article. So, my main recommendation is to reformat the manuscript into a Short communication, as overall the issue is relevant and epidemiologically important. Otherwise, I do not have any major concerns.

Minor comments:

1. Check throughout the manuscript conformity with the template: some headings and pararaphs are wrongly formatted: some References are missing.

2. Check the keywords: they are too long and useless as a search words

3. Check English - pretext use, noun single/plural, 

3. Line 38 - YF is one of the most severe mosquito-borne disease, not hte most - it's argueable

Line 65 - use registered instead of notified

Line 68 - 11 deaths out of 55 confirmed or overall?

Line 82 - please, check - plasma samples, not serum?!

Line 83 - what is IBP?

Line 113 and below - use database name with accession numbers (GenBank Acc.No...)

Line 140 - primary plasma samples?!

Line 165 - the use of term envelope gene is incorrect, it's a genome fragment encoding envelope protein. Genes have starts/stops, so technically, ORFs are genes.

Author Response

The manuscript "Continuous circulation of yellow fever among rural populations in the Central African Republic" present a case of isolation and genetic description of a YFV strain in 2018. The work is nicely performed and presented. Nevertheless, the manuscript describes only one strain and its sequence analysis, which, in my opinion, is not enough for a full-type Original Article. So, my main recommendation is to reformat the manuscript into a Short communication, as overall the issue is relevant and epidemiologically important. Otherwise, I do not have any major concerns.

Thank you for this comment. We could have done a short communication since we are only presenting one yellow fever case. However, many analyses have been performed and would not fit the word limitation of a short communication.

Minor comments:

  1. Check throughout the manuscript conformity with the template: some headings and pararaphs are wrongly formatted: some References are missing.

Response: thank you for this comment. Modifications have been made accordingly

  1. Check the keywords: they are too long and useless as a search words

Response: thank you for this comment. The keywords have been edited for more appropriate ones. Please see line 34-35

  1. Check English - pretext use, noun single/plural, 

Response: Thank you for this comment which has been taking into consideration.

  1. Line 38 - YF is one of the most severe mosquito-borne disease, not the most - it's argueable.

Thank you for this comment. Please see line 41

Line 65 - use registered instead of notified. Please see line 68

Line 68 - 11 deaths out of 55 confirmed or overall?

Thank you for this comment. Overall, 55 YFV cases were laboratory confirmed among which 11 deaths, but 3320 cases were suspected. Please see reference 12 and 13.

Line 82 - please, check - plasma samples, not serum?!

Thank you for this comment. Indeed, it is plasma samples as blood was collected in EDTA tubes

Line 83 - what is IBP?

Thank you for this comment. IBP stands for Institut Pasteur of Bangui. Please see line 86

Line 113 and below - use database name with accession numbers (GenBank Acc.No...)

Thank you for this comment. It has been considered throughout the text.

Line 140 - primary plasma samples?!

Thank you for this comment. Please see line 148

Line 165 - the use of term envelope gene is incorrect, it's a genome fragment encoding envelope protein. Genes have starts/stops, so technically, ORFs are genes.

Thank you for this comment. Please see line 193

Reviewer 3 Report

TCHETGNA et al. describes a new YFV sequence belonging to east-central african genotype. As quite a few YFV whole genomes are avaiable,  specially from the African continent, these findings are very important, as they bring new information regarding virus evolution and maintenance in this continent. However, the article need some improvements before its publication.

Within the methods section, authors should include how many samples were collected and tested by molecular methods, and also how were they kept and sent to the laboratory. I also didn't understand what they've considered as inconclusive results. Did you perform a RT-qPCR for endogenous control? Also, please include ethics concerning mice isolation, including how were they kept and biosafety. Did the authors see clinical manifestations? which ones? How were they euthanized? Did the authors check for cytopatic effect on vero e6 cells or plaque formation?

 Regarding NGS methods, please include the quality of the Illumina sequencing.

I think the results findings are quite interesting, as this sequence is basal that samples collected decades ago, and I suggest authors to include a table showing the nt similatirities and the aa mutations they've found. More, authors states in the discussion section viral rates evolution. As they have performed a bayesian analysis, I believe that they could show more data regarding these findings, and not only the TMRCA, as little is known regarding YFV evolution in its african sylvatic cycle. 

Some minor reviews:

Line 82: insert the proper reference.

Line 253: YFV was not really introduced in China, as only imported cases were detected. I would change this sentence.

Also, check the formation of the results.

Author Response

Point 1: Within the methods section, authors should include how many samples were collected and tested by molecular methods, and also how were they kept and sent to the laboratory. I also didn't understand what they've considered as inconclusive results. Did you perform a RT-qPCR for endogenous control? Also, please include ethics concerning mice isolation, including how were they kept and biosafety. Did the authors see clinical manifestations? which ones? How were they euthanized? Did the authors check for cytopatic effect on vero e6 cells or plaque formation?

Thank you for this comment. Overall, 9 samples were collected from 9 hemorrhagic fever suspected cases in Mboki. (See line 85). Blood samples were transported cool in UN3373 package. Internal control and positive controls are included in the commercial kits used for yellow fever virus detection (RealStar Yellow Fever RT-PCR Kit v1.0 and RealStar Filovirus Screen RT-PCR Kit v1.0, https://www.altona-diagnostics.com/en/products/reagents-140/reagents/realstar-real-time-pcr-reagents/realstar-yellow-fever-virus-rt-pcr-kit-ce.html ). Indeed, the first tests performed on primary plasma sample were all negative. That is why we say inconclusive because we were not able to conclude on the infectious agent. To avoid confusion, Inconclusive has been replaced by negative (See line 149). Viral isolation using mice was performed in a BSL2/3 animal house, the infected mice were kept in an isolated room in sealed ventilated cages. Clinical manifestations observed during the virus isolation on mice include lethargy, bristle fur, paralysis, and death (See line 97). Alive mice were euthanised using a volatile anaesthesia (formol). During virus isolation on Vero E6 cells, only the cytopathic effect was assessed. Our experimental design do not allow the formation of plaque of lysis (See line 99).

  1. Regarding NGS methods, please include the quality of the Illumina sequencing.

Thank you for this comment. The reads obtained after sequencing had a Phred score between 20 and 30

  1. think the results findings are quite interesting, as this sequence is basal that samples collected decades ago, and I suggest authors to include a table showing the nt similatirities and the aa mutations they've found. More, authors states in the discussion section viral rates evolution. As they have performed a bayesian analysis, I believe that they could show more data regarding these findings, and not only the TMRCA, as little is known regarding YFV evolution in its african sylvatic cycle. 

Thank you for this comment. Figure 2 showing the amino acid mutations and percentage of similarities has been added to the manuscript. For the Bayesian analysis, we have only included the TMRCA because it is the most relevant information for us. If we had added other data, it would have not been different from what is already known as not many sequences are available in Central Africa and the Central African Republic.

Line 82: insert the proper reference.

Thank you for this comment. The reference was added. Please see line 85

Line 253: YFV was not really introduced in China, as only imported cases were detected. I would change this sentence.

Thank you for this comment. Effectively, no local transmission was seen in China. The sentence has been reformulated. Please, see line 276

  1. Also, check the formation of the results.

Thank you for this comment

Round 2

Reviewer 1 Report

Improvement seen in this version. The references I have previously suggested should be added to references as there are not many references used in this paper.

Author Response

  1. Improvement seen in this version. The references I have previously suggested should be added to references as there are not many references used in this paper.

Thank you for this comment. We have added previously suggested references (see reference 12, 35 and 36)

Reviewer 3 Report

The authors have provided all information I've requested. However, figure 2 is upside down.

Also, regarding mice euthanasia: formol is not indicated for this, once is higly toxic. Are you sure that formol was used?

Author Response

  1. The authors have provided all information I've requested. However, figure 2 is upside down.

Thank you for this comment. Figure 2 has been well oriented.

  1. Also, regarding mice euthanasia: formol is not indicated for this, once is highly toxic. Are you sure that formol was used?

Thank you for this comment. It was a mistake. Indeed, we combined chloroform and cool temperature for euthanasia.